# Effects of sake lees intake on fecal uremic toxins, plasma D-alanine, constipation, and gut microbiome in healthy adults: A single-arm clinical trial

Toshiaki Tokumaru[1,2], Tadashi Toyama[3], Yusuke Nakade[1], Hisayuki Ogura[1], Megumi Oshima[1], Shiori Nakagawa[1], Masashi Mita[4], Taro Miyagawa[1], Shinji Kitajima[1], Akinori Hara[5], Norihiko Sakai[1], Miho Shimizu[1], Yasunori Iwata[1]*, Takashi Wada[1]

1 Department of Nephrology and Rheumatology, Kanazawa University, Kanazawa, Japan, 2 Department of Nutrition Management, Kanazawa University Hospital, Kanazawa, Japan, 3 Department of Nephrology, Faculty of Medical Sciences, University of Fukui, Fukui, Japan, 4 KAGAMI INC., Osaka, Japan, 5 Department of Hygiene and Public Health, Kanazawa University, Kanazawa, Japan

* iwatay@staff.kanazawa-u.ac.jp

## Abstract

### Background

Sake lees consumption has the potential to reduce uremic toxins by influencing the gut microbiome. To lay the groundwork for a clinical trial targeting chronic kidney disease (CKD) patients, we conducted a pilot study to explore the relationship between sake lees intake and changes in fecal uremic toxin levels among individuals with constipation. D-alanine, a renoprotective component of sake lees, was also evaluated.

### Methods

This single-arm, before and after study lacked a control group. Participants met the diagnostic criteria for chronic constipation. They consumed 25 or 50 g of sake lees daily for 6 weeks. The primary endpoint was the change in fecal indole levels from baseline. Secondary endpoints included changes in plasma D-alanine, Constipation Scoring System (CSS) scores, and the composition of the fecal microbiome.

### Results

Eight participants, with a mean age of 46 years, completed the study. Percentage changes in fecal indole levels were +42%, +52%, and −6% at weeks 2, 4, and 6, respectively. Plasma D-alanine levels showed percentage changes of +39%, +24%, and +38% at the same time points. CSS scores improved from 9.2 to 6.8 by week 2 and remained stable after week 4. The proportion of the phylum Firmicutes in the gut microbiome increased slightly from 53% at baseline to 57% by week 6.

**Data availability statement:** The minimal dataset underlying the findings of this study is available from the Figshare repository at the following DOI: https://doi.org/10.6084/m9.figshare.28755158.

**Funding:** Initials of the authors who received each award: T.Tok Grant numbers awarded to each author: JP21lk0310074 The full name of each funder: Japan Agency for Medical Research and Development (AMED) URL of each funder website: https://www.amed.go.jp/en/index.html Funder's roles: Funding and advice for the overall research The funders had no role in study design, data collection and analysis, decision to publish, or preparation of the manuscript.

**Competing interests:** The authors have declared that no competing interests exist.

## Conclusion

Sake lees intake may reduce fecal uremic toxins, elevate plasma D-alanine levels, alleviate constipation, and modify the gut microbiome. However, future studies are needed to confirm these effects in patients with CKD.

## Introduction

Microbiome-targeted treatments for chronic kidney disease (CKD) are increasingly gaining attention [1]. Patients with CKD often experience gut microbiome dysbiosis, characterized by an imbalance in microbial composition [2]. This dysbiosis contributes to elevated intestinal uremic toxin levels, which can accelerate CKD progression [3]. Additionally, constipation, a common symptom in patients with CKD, is also associated with dysbiosis [4]. The laxative linaclotide has shown potential in slowing CKD progression and reducing the risk of cardiovascular disease [5]. Probiotics and prebiotics are emerging as key therapeutic options. Certain probiotics or prebiotics have been found to improve gut microbiome health in patients with CKD and lower blood uremic toxin levels or blood urea nitrogen [6,7].

Sake lees, a fermented food widely consumed in Japan [8], are a byproduct of the rice wine (sake) fermentation process. They are rich in dietary fiber, and daily consumption of 50 g has been shown to alleviate constipation in the general population [9]. Sake lees have low levels of phosphorus and potassium and are free of salt, making them a suitable option for patients with CKD. Furthermore, fermented foods like sake lees are notable for their high D-amino acid content, which is produced by microorganisms during fermentation [10].

Certain bacteria produce D-alanine and D-serine, both associated with renoprotective properties [11,12]. Sake lees contains various bacteria, including those capable of producing these compounds. Consequently, sake lees are anticipated to lower uremic toxin levels by alleviating constipation and promoting kidney protection through the actions of D-alanine and D-serine.

To date, no clinical trials have investigated the use of sake lees in patients with CKD. Before initiating such trials, it is essential to assess the effects of sake lees on uremic toxin levels and determine an effective dosage in non-CKD individuals. Additionally, understanding variations in D-alanine and D-serine levels is of interest. This pilot study aimed to evaluate the impact of sake lees intake on fecal uremic toxins in non-CKD individuals with chronic constipation, a condition commonly associated with CKD. Changes in plasma D-alanine, D-serine, constipation severity, and the gut microbiome were also examined.

## Materials and methods

### Study design and participants

This pilot study investigated the effects of sake lees intake on fecal uremic toxin levels in individuals with chronic constipation. It employed an open-label, single-arm, before and after design without a control group. No previous studies have

investigated changes in fecal uremic toxins in patients with CKD after sake lees intake. Therefore, evaluating whether sake lees intake changes fecal uremic toxins in non-CKD patients is necessary. Because this was an exploratory study, a single-arm study design was considerably appropriate based on ethical considerations, and the study was conducted without a control group [13]. A sample size estimation was not performed due to the absence of prior data on effect size. An attempt was made to determine the sample size using hypothetical effect size; however, the calculated sample size was not feasible for research concerning budget and time period. Participants were eligible if they were aged 20 years or older and met the diagnostic criteria for chronic constipation [14]. Exclusion criteria included alcohol intolerance, antibiotic use within the previous 4 weeks, or use of medications causing diarrhea or constipation. Participant recruitment was carried out through Kanazawa University notice boards and mailing lists, targeting Kanazawa University employees and their families. The study was conducted between December 2022 and March 2023. The protocol for this trial and supporting TREND statement checklist are available as supporting information (**S1** Protocol and **S1** TREND Checklist).

## Sake lees used in this study

Sake lees are the white solid residue remaining after fermenting a mixture of rice, rice malt, and water with yeast. They are a byproduct of Japanese sake production and have been a traditional food for a long time. The sake lees used in this study were sourced from the Japanese sake "Kakuma no Sato" and purchased directly from the manufacturer in Japan. According to the manufacturer's specifications, the nutritional content per 100 g of sake lees was as follows: energy, 209 kcal; protein, 7.2 g; fat, 0.6 g; carbohydrate, 30.8 g; salt, 0 g; and dietary fiber, 5.2 g. The D-amino acid content was measured at 12 μmol of D-alanine and 0.7 μmol of D-serine per 100 g of sake lees.

## Dietary treatment

The daily sake lees intake amounts were set at 25 g and 50 g, based on a previous study [9]. Participants were randomly assigned to either 25 g/day or 50 g/day of sake lees. A 2-week run-in period was implemented to minimize the impact of food or medications on the intestinal environment, followed by a 6-week dietary treatment period with sake lees. These timeframes were chosen for feasibility. During the run-in period, sake lees intake was prohibited. Participants were not allowed to consume functional foods or medications marketed during the run-in and treatment periods to improve gut health. However, consuming yogurt or other common fermented foods was permitted during both periods. Since fecal indole and p-cresol are metabolites produced by gut bacteria from dietary protein, participants were instructed to maintain their usual eating habits throughout the run-in and treatment periods. Additionally, they were instructed to maintain their usual fluid intake and physical activity habits considering these can influence constipation.

The same batch of sake lees was used throughout the study and stored in the refrigerator during the study period. The daily amount of sake lees to be consumed was assigned to each participant. Participants were free to decide when and how much sake lees to consume per meal, with the option to take the full 50 g in one meal or split it into smaller portions. A recipe was provided to help participants cook the sake lees, though they were also allowed to prepare it in their preferred way. Incentives (gift certificates) were provided to increase the adherence of participants. The allocation of sake lees intake and information on the behavior of participants during the study period were provided by the registered dietician at Kanazawa University Hospital.

## Primary and secondary endpoints

The primary endpoints were the percentage changes in fecal indole and p-cresol levels from baseline. The secondary endpoints were the percentage changes in the plasma D-alanine and D-serine levels, the Constipation Scoring System (CSS) score [15], the Patient Assessment of Constipation Quality of Life (PAC-QOL) score [16], the fecal microbiome composition at the phylum level, the proportion of *Bacteroides* (indole-producing bacteria) [17], and the proportion of Coriobacteriaceae (p-cresol-producing bacteria) [18]. Additionally, alpha diversity (number of operational taxonomic units

[OTUs], Chao-1 index, and Shannon index), beta diversity (unweighted and weighted UniFrac distance), and adherence to sake lees intake were assessed. Regarding blood uremic toxins, fecal and blood uremic toxins correlate in patients with CKD, and lowering fecal uremic toxins is a treatment strategy [1]. However, the participants were non-CKD patients, and the levels of blood uremic toxins, such as indoxyl sulfate, would not change, potentially because of excretion from the kidneys. Therefore, the blood uremic toxin levels were not measured.

## Sample collection

Samples were collected at baseline, week 2, week 4, and week 6. Participants' questionnaires provided information on age, sex, CSS, and PAC-QOL scores. Body weight was measured using calibrated scales. At baseline and week 6, nutritional intake data (energy, protein, salt, and dietary fiber) were gathered through a food frequency questionnaire based on food groups [19]. Participants recorded their daily sake lees intake and noted any prohibited food or drugs consumed on a daily log. Fecal samples for indole and p-cresol analysis were collected by participants in a stool container using a specimen-collection spoon and then stored at −80°C. Fecal samples for gut microbiome analysis were taken using a brush kit (TechnoSuruga Laboratory Co., Ltd., Shizuoka, Japan). Blood was drawn into tubes containing anticoagulant, and plasma was separated by centrifugation and stored at −80°C. Participants consumed a portion-controlled meal 24 h before providing the blood sample. Detailed information about the portion-controlled meal is provided in S1 Fig.

## Determination of fecal indole and p-cresol levels

The levels of fecal indole and p-cresol were analyzed by Techno Suruga Laboratory Co., Ltd. using gas chromatography/mass spectrometry (Shizuoka, Japan). Additional details on the analysis of these fecal metabolites can be found in S1 File.

## Determination of chiral amino acids by two-dimensional high-performance liquid chromatography

As previously described, D- and L-amino acids were analyzed using a two-dimensional (2D) high-performance liquid chromatography system (NANOSPACE SI-2 series, Shiseido, Tokyo, Japan) [20,21]. In brief, 4-nitrobenzo-2-oxa-1,3-diazole derivatives of the amino acids (NBD-AAs) were first separated in the first dimension using an online fraction collector with a microbore-ODS column ($1,000 \times 0.53$ mm$^2$ i.d., 45°C; Shiseido). The isolated fractions were then automatically transferred to a second dimension, where a narrow-bore enantioselective column (KSAACSP-001S, $250 \times 1.5$ mm$^2$ i.d., 25°C; prepared in collaboration with Shiseido) was used to identify the D- and L-enantiomers. The mobile phase in the second dimension consisted of methanol and acetonitrile mixed with formic acid. Fluorescence detection of the NBD-AAs was performed at 530 nm, with excitation at 470 nm.

## DNA extraction and sequencing

DNA was analyzed using 16S ribosomal ribonucleic acid (rRNA) sequencing at Takara Bio's Biomedical Center (Shiga, Japan). Genomic DNA was extracted from the samples using the NucleoSpin Microbial DNA (Macherey-Nagel, Duren). The primers used were 341F
(5′-TCGTCGGCAGCGTCAGATGTGTATAAGAGACAGCCTACGGGNGGCWGCAG-3′) and 806R
(5′-GTCTCGTGGGCTCGGAGATGTGTATAAGAGACAGGGACTACHVGGGTWTCTAAT-3′), with Illumina adapter overhang sequences. A second polymerase chain reaction amplification was carried out using the Nextera XT Index Kit v2 (Illumina, San Diego, USA). Sequencing libraries were purified with the Agencourt AMPure XP (Beckman Colter) and quantified by fluorescence using the Quant-iT dsDNA Assay Kit (Thermo Fisher Scientific, Massachusetts, USA). Clonal clusters of the libraries were generated and sequenced in $2 \times 250$-bp mode on the MiSeq system (Illumina) using the MiSeq Reagent v3 kit.

## 16S rRNA analysis

The analysis was conducted using QIIME2 (version 2021.2). Sequences were demultiplexed based on per sample barcodes, and Illumina-sequenced amplicon read errors were corrected using DADA2, followed by clustering into OTUs at 99% identity with VSEARCH [22]. Taxonomy classification was performed using the Greengenes 99% OTU database (version 13_8). Changes in bacterial proportions were analyzed at the phylum level, while relative abundance comparisons were made at the class, family, order, genus, and species levels before and after sake lees intake. A rarefaction curve was generated after random sampling of up to 50,000 sequences per sample. Alpha diversity metrics (number of OTUs, Chao-1 index, and Shannon index) were calculated for each sample at the sampling depth and compared before and after sake lees intake. Principal coordinate analysis was used to assess beta diversity by calculating the weighted and unweighted UniFrac distances across samples, with a sampling depth of 10,000 sequences.

## Statistical analysis

Baseline data with a normal distribution were presented as mean and standard deviation (SD). Categorical variables were reported as frequencies and proportions. Percentage changes in fecal indole, fecal p-cresol, plasma D-alanine, and plasma D-serine levels from baseline were calculated and assessed using a one-sample t-test with a test value of 0. $p$-values were adjusted using the Bonferroni method. Changes in the CSS and PAC-QOL scores were evaluated using a paired t-test for the total score. For PAC-QOL, values for questions 18 and 25–28 were reversed and added to the scores of the other questions. The number of OTUs, Chao-1 index, Shannon index, and proportions of *Bacteroides* and Coriobacteriaceae were analyzed using repeated measures analysis of variance. A two-tailed significance level of $p < 0.05$ was applied. The full analysis set (FAS) and per protocol set (PPS) were used for statistical analysis. All analyses were performed using Stata/MP statistical software (version 17; StataCorp LLC, College Station, TX, USA).

## Ethics approval

The study protocol was approved by the Ethics Committee of Kanazawa University Hospital (Approval No.: 2021-222-113867) and was conducted in accordance with the Declaration of Helsinki. This study was registered on the University Hospital Medical Information Network Clinical Trials Registry (UMIN-CTR [UMIN000056444]) on December 18, 2024. This pilot study was conducted as a preliminary investigation while creating a clinical trial protocol for patients with CKD. Although ethical approval had been obtained from the ethics committee prior to the study, registration with the UMIN-CTR had not yet been completed at the time of trial initiation. The authors confirm that all ongoing and related trials for this intervention are registered. Since the UMIN-CTR accepts registrations after the clinical trial has ended, the trial was registered after its completion. Such late registration does not affect the validity of study results or the safety and rights of participants.

## Results

### Baseline characteristics of participants

Fig 1 shows the participant selection process. Eight participants who met the inclusion criteria were enrolled in the study. After randomization, four participants were assigned to consume either 25 g/day or 50 g/day of sake lees. All participants completed both the run-in and sake lees intake periods without any protocol violations. Both the FAS and the PPS included all eight participants. Compliance with the sake lees intake was 100% for both the 25 g and 50 g groups throughout the dietary period. Moreover, none of the participants consumed prohibited foods or medications, and there were no side effects such as diarrhea or constipation.

Table 1 depicts the baseline characteristics of the study participants. The average age was 46 years, with 75% of participants being women. The average levels of fecal indole and p-cresol were 45 µg/g and 130 µg/g, respectively. The

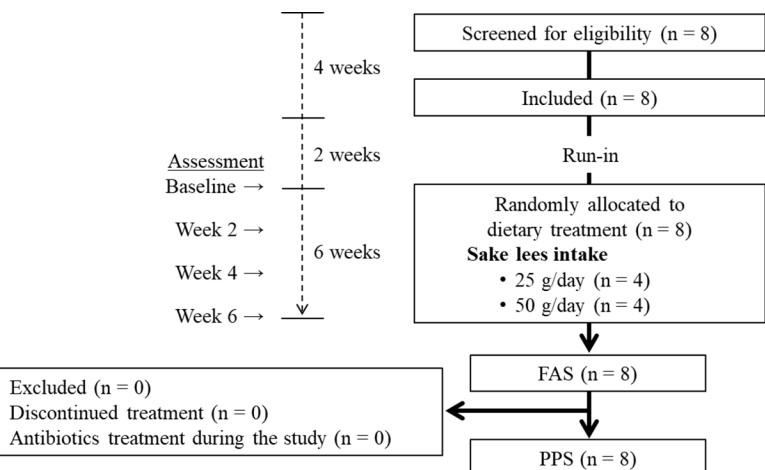

**Fig 1. Flow diagram illustrating the participant selection process.** FAS, full analysis set; PPS, per protocol set.

**Table 1. Baseline characteristics of the study participants.**

| Variables | Overall (n=8) | | 25 g sake lees (n=4) | | 50 g sake lees (n=4) | |
|---|---|---|---|---|---|---|
| Age, years | 46 | (9) | 44 | (4) | 49 | (12) |
| Female, n (%) | 6 | (75) | 3 | (75) | 3 | (75) |
| Body mass index, kg/m² | 19.7 | (2.7) | 18.7 | (0.7) | 20.7 | (3.7) |
| Fecal indole, µg/g | 45 | (26) | 39 | (33) | 52 | (20) |
| Fecal p-cresol, µg/g | 130 | (27) | 118 | (39) | 140 | (35) |
| Plasma D-alanine, nmol/mL | 0.89 | (0.36) | 0.72 | (0.19) | 1.06 | (0.43) |
| Plasma D-serine, nmol/mL | 2.22 | (0.80) | 1.84 | (0.25) | 2.60 | (1.02) |
| CSS score, points | 9.3 | (3.0) | 8.6 | (1.3) | 9.8 | (4.3) |
| PAC-QOL score, points | 43 | (15) | 40 | (15) | 46 | (18) |
| Nutritional intake | | | | | | |
| Energy, kcal/day | 1,784 | (332) | 1,804 | (282) | 1,764 | (422) |
| Protein, g/day | 64 | (17) | 56 | (11) | 71 | (20) |
| Salt, g/day | 9.2 | (3.1) | 7.6 | (0.6) | 10.9 | (3.8) |
| Dietary fiber, g/day | 12.5 | (3.7) | 11.1 | (1.3) | 13.9 | (5.0) |

Data are presented as number (%) or mean (standard deviation). CSS, Constipation Scoring System; PAC-QOL, Patient Assessment of Constipation Quality of Life.

mean plasma D-alanine and D-serine levels were 0.89 nmol/mL and 2.22 nmol/mL, respectively. The mean CSS score was 9.3 and the average PAC-QOL score was 43.

## Fecal indole and p-cresol changes

**Fig 2** shows the changes in fecal uremic toxin levels. The mean percentage change from baseline for fecal indole was +42% at week 2, +52% at week 4, and −6% at week 6 (**Fig 2A**). No significant relationship was found with sake lees intake. For fecal p-cresol, the mean percentage change from baseline was +7% at week 2, −20% at week 4, and −20% at week 6 (**Fig 2B**). No significant relationship was observed with sake lees intake. Changes in fecal indole and p-cresol levels based on sake lees intake amounts are shown in S1 Table.

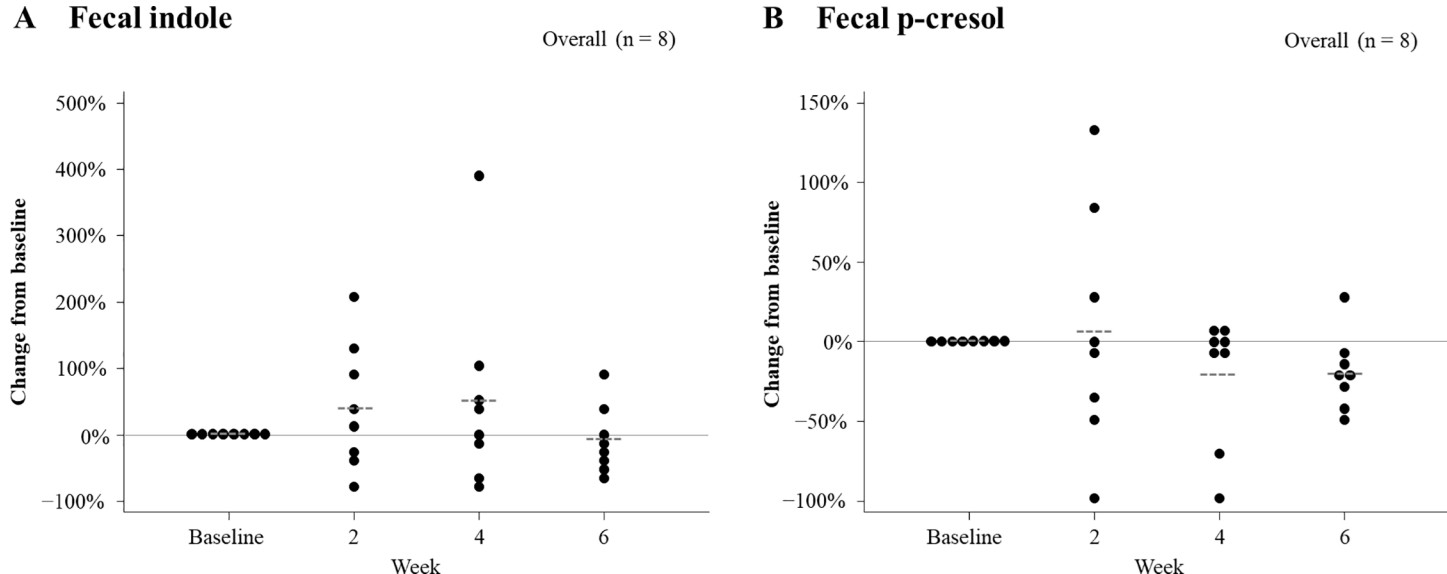

**Fig 2. Changes in fecal uremic toxin levels.** The changes in fecal indole (A) and p-cresol (B) levels from baseline to 6 weeks with sake lees intake. Data are shown as percentage changes from baseline values. The dotted lines represent the mean values. No significant association was observed for (A) and (B).

### Plasma D-alanine and D-serine changes

**Fig 3** shows the changes in plasma D-alanine and D-serine levels. The mean percentage change from baseline for plasma D-alanine was +39% at week 2, +24% at week 4, and +38% at week 6, with a significant increase at week 2 ($p=0.04$). The change was more pronounced with 50 g of sake lees compared to 25 g (**Fig 3A**). The mean percentage change for plasma D-serine was +4% at week 2, −7% at week 4, and −8% at week 6, with no significant association observed (**Fig 3B**). The changes in plasma D-alanine and D-serine levels by sake lees intake are detailed in S1 Table.

### CSS and PAC-QOL score changes

**Fig 4** shows the changes in the CSS and PAC-QOL scores. The CSS score at week 2 was 6.8, which represented a significant decrease compared to the baseline score ($p<0.001$) and continued through week 6 (**Fig 4A**). Similarly, the mean PAC-QOL score at week 2 was 25, showing a significant reduction from baseline ($p<0.001$), which continued through week 6 (**Fig 4B**). The changes in CSS and PAC-QOL scores based on sake lees intake are provided in S1 Table.

### Fecal microbiome changes

**Fig 5** shows the changes in the fecal microbiome at the phylum level. The dominant phylum was Firmicutes (53%), followed by Actinobacteria (21%), Bacteroidetes (17%), Verrucomicrobia (6%), and Proteobacteria (2%). Firmicutes increased to 56% at week 2, 55% at week 4, and 57% at week 6. Verrucomicrobia decreased to 4% at week 2, 2% at week 4, and 1% at week 6.

Changes in the proportion of the genus *Bacteroides* and the family Coriobacteriaceae are shown in S2 Fig. No significant associations were found between *Bacteroides* and sake lees intake ($p=0.87$) or between Coriobacteriaceae and sake lees intake ($p=0.85$).

Alpha diversity, which reflects the number of species within a single sample, did not show changes in the number of OTUs ($p=0.79$), Chao-1 index ($p=0.81$), and Shannon index ($p=0.30$) before and after sake lees intake (S3 Fig). S4 Fig

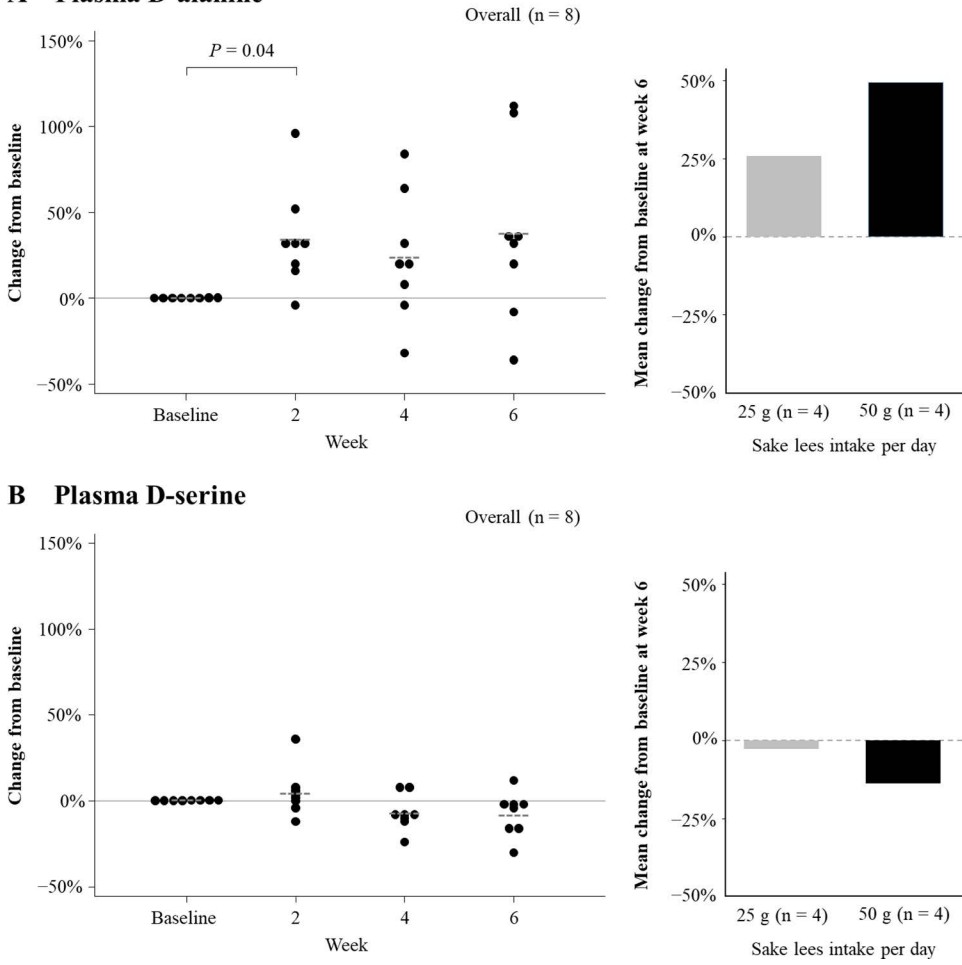

**Fig 3. Changes in plasma D-alanine and D-serine levels.** The changes in plasma D-alanine (A) and D-serine (B) levels are shown. On the left, the percentage change from baseline is presented. The dotted lines represent the mean values. On the right, the mean change from baseline for D-alanine and D-serine at week 6 is shown for 25 g or 50 g daily sake lees intake. No significant association was observed for (B).

shows beta diversity, which indicates environmental differences between samples. The unweighted UniFrac distance did not show significant changes from baseline to week 6, while the weighted UniFrac distance did change from baseline to week 6 compared to the unweighted UniFrac distance.

## Nutritional intakes

Nutrient intake, excluding sake lees, was compared between baseline and after 6 weeks, and no changes were observed in energy, protein, salt, or dietary fiber (S1 Table). There was also no change in the frequency or quantity of protein-rich foods such as fish and meat.

## Discussion

This study was a single-arm, before and after pilot study involving eight individuals with chronic constipation, serving as a preliminary step for a clinical trial in patients with CKD. After consuming 25 g or 50 g of sake lees daily for 6 weeks, changes in fecal uremic toxin levels were observed. These consumption patterns also influenced plasma D-amino acid

**A   Constipation Scoring System**

**B   PAQ-QOL**

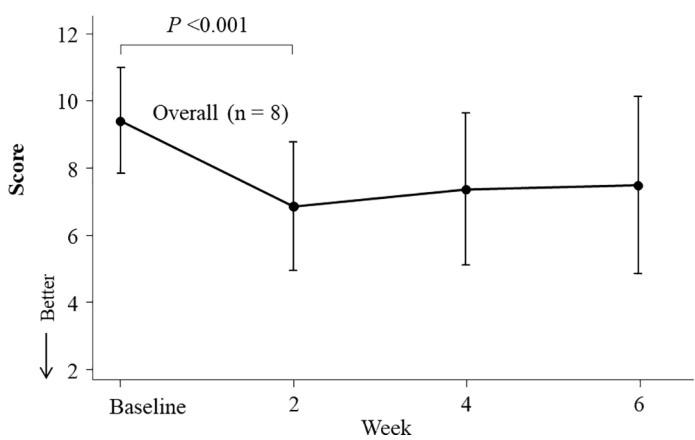
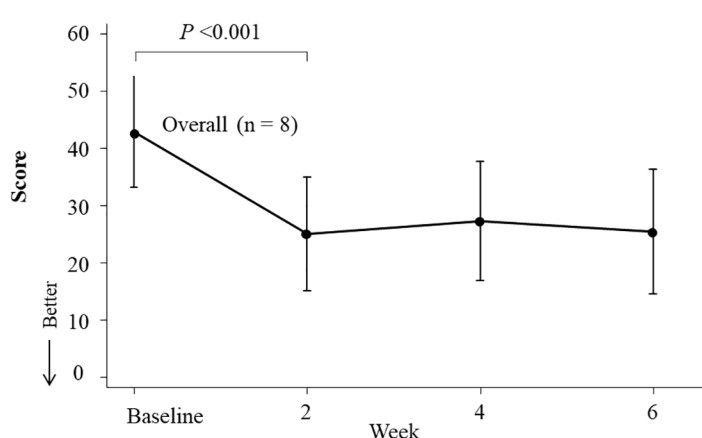

**Fig 4. Changes in the CSS and PAC-QOL scores.** The CSS score (A) and PAQ-QOL score (B) are shown. Data represent the mean of eight values with a 95% confidence interval.

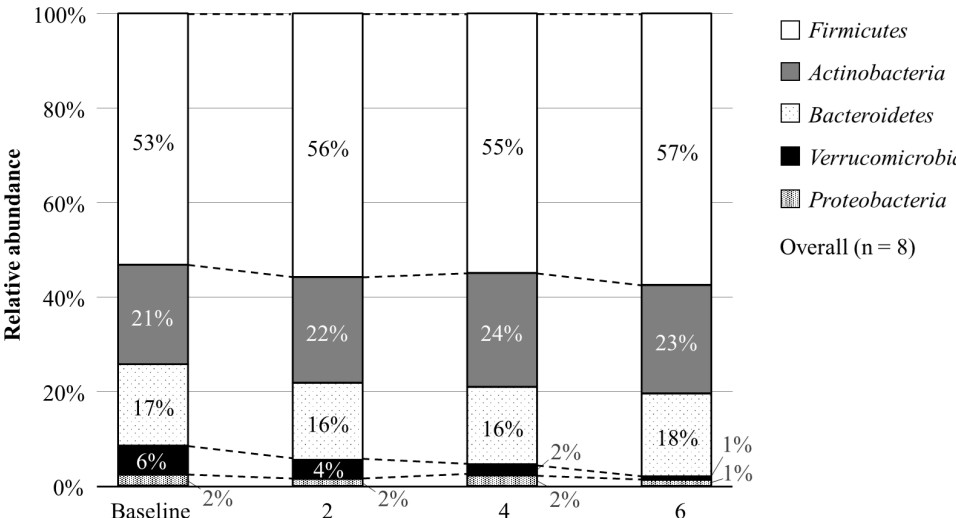

Legend:
- □ *Firmicutes*
- ▨ *Actinobacteria*
- □ *Bacteroidetes*
- ■ *Verrucomicrobia*
- ▦ *Proteobacteria*

Overall (n = 8)

**Fig 5. Changes in the fecal microbiome at the phylum level.** The percentages at each time point may not sum to 100% due to rounding.

levels, constipation status, and the fecal microbiome. Although these findings indicate the therapeutic effects of sake lees, further studies with control group and sufficient sample size are needed for confirmation.

Fecal indole and p-cresol levels initially increased at week 2 but gradually decreased by weeks 4 and 6. This decline may be related to the improvement of constipation. The laxative linaclotide has contributed to the reduction of trimethylamine N-oxide, which is one of the uremic toxins [5]. Alternatively, this decline may be due to a sustained improvement in the bacteria responsible for producing indole or p-cresol, which could potentially explain the reduced levels of these substances [23]. For instance, *Bacteroides* and *Clostridium* are known to produce indole [17], while Coriobacteriaceae and Enterobacteriaceae are associated with p-cresol production [18]. However, no significant changes in the levels of these

indole- or p-cresol-producing bacteria were observed in this study. It is possible that bacteria other than those investigated in this study are involved, and this is a topic for future study.

In this study, plasma D-alanine levels increased with sake lees intake, while plasma D-serine levels remained unchanged. Since sake lees contain more than 10 times as much D-alanine as D-serine, it is likely that D-alanine was absorbed, but D-serine did not show a change due to its lower content. Previous studies have indicated that plasma D-alanine levels are negatively correlated with estimated glomerular filtration rate and may have a protective effect on the kidneys [11,12]. Therefore, it is possible that sake less intake could further increase plasma D-alanine levels in patients with CKD. However, as this study was conducted on individuals without kidney disease, further studies involving patients with CKD are needed to confirm these potential benefits.

Constipation improved after consuming sake lees in this study. Previous studies have found that sake lees consumption increased defecation frequency and improved stool characteristics [8,9,24]. Watanabe et al. suggested a 50 g/day of intake of sake lees to alleviate constipation [9]. Additionally, resistant proteins, which are not digested in the intestine, may help with constipation [9]. Other research has indicated that the dietary fiber and oligosaccharides in sake lees could also benefit constipation [25]. The lack of differences in the CSS and PAC-QOL scores based on sake lees amount is unclear, but it may be due to the short duration of the dietary treatment in this study. Because this was an open-label, single-arm study, changes in the participants' usual habits, including diet, fluid intake, and physical activity, which can influence constipation, might have occurred because of information bias. Additionally, because the timing of sake lees intake was not fixed to ease implementation, this might have influenced the results.

In this study, sake lees intake changed the phylum proportions of the fecal microbiome, which aligns with previous research indicating that fermented foods, such as sake lees, impact the gut microbiome [26,27]. For instance, an increase in the Firmicutes phylum, which is considered beneficial for health, was observed. One possible explanation for this change could be the dietary fiber content in sake lees [28]. Sake lees is a fermented food that contains a variety of bacteria and may contribute to changes in the fecal microbiome [26,27]. Although this study was a 6-week intervention, our findings indicate the possibility that longer-term interventions can sustain changes in the gut microbiome and provide lasting health benefits.

Long-term commitment is required for dietary treatments, so clinical studies must assess adherence and potential side effects. In this study, both the 25-g and 50-g sake lees intake groups showed good adherence. Participants consumed sake lees based on their preference. Standardizing consumption methods and timing may improve outcome interpretation. No adverse events such as diarrhea or constipation were reported by participants. Previous studies have also found no adverse events linked to sake lees intake [24]. In contrast, current drug treatments targeting intestinal uremic toxins in patients with CKD often face low compliance [29]. If sake lees dietary treatment proves effective for patients with CKD, it could complement these drug treatments.

This study has several limitations. First, this was a pilot study that used a single-arm design without a control group, potentially limiting the interpretation. This study was conducted as an early phase of clinical trial implementation for patients with CKD. The primary objectives were to evaluate the feasibility, collect sample size information, and assess the potential effects of sake lees intake on the selected biomarkers. In this early exploratory phase, a single-arm design was used for feasibility and ethical considerations. Second, as an open-label study, the CSS and PAC-QOL scores, which were evaluated by participants using subjective measures, may have been influenced by expectancy bias (a type of information bias). Third, we could not determine the sample size based on effect size or statistical power before the study owing to feasibility constraints. The small sample size may have reduced the statistical power and increased the likelihood of distortion due to outliers or random fluctuations. Furthermore, the participants do not accurately reflect the population, potentially limiting the generalizability of the findings. Post hoc power analysis revealed that, assuming a difference of 50 μg/g in fecal indole (approximately half of the 112 μg/g difference reported in a previous randomized controlled trial [30]) and extrapolating the SD (30 μg/g), our actual sample size of the clinical trials in patients with CKD (n = 11 and 13) would yield a statistical power of 97%. Fourth, the impact of the timing of sake lees intake or the amount of sake lees consumed

at one time on the results was not explored. Fifth, factors influencing constipation improvements, such as fluid intake and physical activity, were not considered in this study. Sixth, safety evaluation could not be conducted because of the absence of a control group. Sake lees have been consumed in Japan for centuries, and its safety has been confirmed in trials involving healthy individuals [8]. In the next clinical trial targeting patients with CKD, a control group must be included to assess safety. Lastly, changes in fecal uremic toxins were used as a surrogate for blood uremic toxins, which may not fully reflect changes in blood levels.

Sake lees are beneficial for the dietary treatment of patients with CKD because they are low in phosphorus and potassium and contain no salt. However, it is unclear whether sake lees intake in patients with CKD effectively reduces uremic toxins, improves constipation, and alters the gut microbiome. Further clinical trials are needed to confirm their efficacy.

## Conclusions

Sake lees intake may reduce fecal uremic toxins, increase plasma D-alanine levels, improve constipation, and alter the gut microbiome. Although these results are considered meaningful, caution is needed in interpreting the therapeutic effects. Therefore, further studies with a control group are needed to assess the impact of sake lees on uremic toxins and the gut microbiome in patients with CKD.

## Supporting information

**S1 Fig. Portion-controlled meal for blood sampling.** The dinner before blood sampling, along with breakfast and lunch on the sampling day, followed the same portion-controlled meal plan. During this period, only water and tea were allowed, with other drinks prohibited. The amount of sake lees consumed on the day before and the day of blood sampling was consistent between the 25-g and 50-g sake lees groups.
(DOCX)

**S2 Fig. Changes in the proportions of indole- and p-cresol-producing bacteria.** The changes in the proportions of *Bacteroides* (indole-producing) (A) and Coriobacteriaceae (p-cresol-producing) (B) are shown. Dotted lines represent the mean values (n = 8).
(DOCX)

**S3 Fig. Alpha diversity indices.** OTUs, operational taxonomic units. Dotted lines indicate the mean values (n = 8).
(DOCX)

**S4 Fig. Principal coordinates analysis of unweighted (A) and weighted UniFrac distances (B) illustrating beta diversity (species level) in the fecal microbiome composition across participants.** The left and right graphs represent the same dataset. On the left, dots of the same color indicate the same participant across different weeks, with the number of weeks represented in grayscale on the right.
(DOCX)

**S1 Table. Changes in endpoints from baseline with sake lees intake.** Data are presented as mean values (SD). Nutritional intake exludes nutrients derived from sake lees. CSS, Constipation Scoring System; PAC-QOL, Patient Assessment of Constipation Quality of Life.
(DOCX)

**S1 Protocol. Sake lees and uremic toxins.**
(DOCX)

**S2 Protocol. Protocol in Japanese.**
(DOCX)

**S1 Checklist. TREND Checklist.**
(DOC)

**S1 File. Measurement of indole, phenol, skatole, and p-cresol.**
(DOCX)

## Acknowledgments

The authors thank the study participants in this study.

## Author contributions

**Conceptualization:** Toshiaki Tokumaru, Tadashi Toyama, Yusuke Nakade, Yasunori Iwata, Takashi Wada.

**Data curation:** Toshiaki Tokumaru, Tadashi Toyama.

**Formal analysis:** Toshiaki Tokumaru, Tadashi Toyama.

**Funding acquisition:** Toshiaki Tokumaru, Yusuke Nakade.

**Investigation:** Toshiaki Tokumaru.

**Methodology:** Toshiaki Tokumaru, Tadashi Toyama, Yusuke Nakade, Yasunori Iwata.

**Project administration:** Takashi Wada.

**Supervision:** Hisayuki Ogura, Yasunori Iwata, Takashi Wada.

**Visualization:** Toshiaki Tokumaru, Tadashi Toyama.

**Writing – original draft:** Toshiaki Tokumaru.

**Writing – review & editing:** Toshiaki Tokumaru, Tadashi Toyama, Yusuke Nakade, Hisayuki Ogura, Megumi Oshima, Shiori Nakagawa, Masashi Mita, Taro Miyagawa, Shinji Kitajima, Akinori Hara, Norihiko Sakai, Miho Shimizu, Yasunori Iwata, Takashi Wada.

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
