## [Decision Letter · Decision Letter 0]

6 Jan 2025

PONE-D-24-55416Effects of sake lees intake on fecal uremic toxins, plasma D-alanine, constipation, and gut microbiome in healthy adults: A single-arm clinical trialPLOS ONE

Dear Dr. Tokumaru,

Thank you for submitting your manuscript to PLOS ONE. After careful consideration, we feel that it has merit but does not fully meet PLOS ONE’s publication criteria as it currently stands. Therefore, we invite you to submit a revised version of the manuscript that addresses the points raised during the review process.

We look forward to receiving your revised manuscript.

Kind regards,

Sayed Haidar Abbas Raza

Academic Editor

PLOS ONE

2. Thank you for submitting your clinical trial to PLOS ONE and for providing the name of the registry and the registration number. The information in the registry entry suggests that your trial was registered after patient recruitment began. PLOS ONE strongly encourages authors to register all trials before recruiting the first participant in a study.

1) your reasons for your delay in registering this study (after enrolment of participants started);

2) confirmation that all related trials are registered by stating: “The authors confirm that all ongoing and related trials for this drug/intervention are registered”.

3. Thank you for stating the following financial disclosure:  [Initials of the authors who received each award: T.Tok

Grant numbers awarded to each author: JP21lk0310074

The full name of each funder: Japan Agency for Medical Research and Development (AMED)

URL of each funder website: https://www.amed.go.jp/en/index.html

Funder's roles: Funding and advice for the overall research].  Please state what role the funders took in the study.  If the funders had no role, please state: "The funders had no role in study design, data collection and analysis, decision to publish, or preparation of the manuscript." If this statement is not correct you must amend it as needed. Please include this amended Role of Funder statement in your cover letter; we will change the online submission form on your behalf.

Additional Editor Comments (if provided):

Reviewers' comments:

Reviewer's Responses to Questions

**Comments to the Author**

1. Is the manuscript technically sound, and do the data support the conclusions?

Reviewer #1: Partly

Reviewer #2: Partly

2. Has the statistical analysis been performed appropriately and rigorously? 

Reviewer #1: No

Reviewer #2: Yes

3. Have the authors made all data underlying the findings in their manuscript fully available?

Reviewer #1: Yes

Reviewer #2: Yes

4. Is the manuscript presented in an intelligible fashion and written in standard English?

Reviewer #1: No

Reviewer #2: Yes

5. Review Comments to the Author

Reviewer #1: This pilot study provides an intriguing look at the potential benefits of sake lees intake in individuals with chronic constipation. The authors present evidence suggesting that daily consumption of sake lees (either 25 g or 50 g) may reduce fecal uremic toxin levels, increase plasma D-alanine levels, improve constipation, and modulate the gut microbiome. These findings, if validated in larger and more rigorous trials, could have important implications for managing constipation in both non-CKD individuals and potentially in CKD patients—particularly due to the low levels of phosphorus, potassium, and sodium in sake lees.

Major Points for Improvement

1. The lack of a control group makes it difficult to attribute observed outcomes solely to sake lees intake. Including a placebo or “no intervention” group would strengthen causal inferences.

2. Because participants were aware of their treatment status, subjective measures (e.g., constipation severity scores) may be influenced by expectancy bias. Blinded designs or at least blinded endpoint assessments would help mitigate this risk.

3. The manuscript acknowledges that the sample size was not determined based on effect size or power calculations. Providing a more robust sample size calculation—based on preliminary data or existing literature—will enhance the credibility of the results.

4. While fecal uremic toxin changes are an interesting surrogate, the manuscript would benefit from clarifying how well they correlate with blood levels or clinical outcomes. It would be worthwhile to measure blood uremic toxin levels or additional relevant biomarkers to strengthen the link between gut environment changes and systemic impacts.

5. Factors such as fluid intake, physical activity, and timing of sake lees consumption could significantly influence constipation and gut microbiome outcomes. These should be explicitly addressed in methods and discussion sections to rule out confounding effects.

Minor Points for Improvement

1. Proofread for minor typos (e.g., “To data” in line 61 should be “To date”), consistent terminology (e.g., “sake lees” vs. “Sake lees”), and uniform tense throughout the manuscript.

2. Ensure that all figures and tables are clearly labeled with descriptive titles and legends, and that axis labels are self-explanatory. Double-check that the numbering of figures and references in the main text aligns with the final layout.

3. Verify that references follow the journal’s style guide and that all in-text citations match the reference list. In particular, older references might be updated with more recent studies addressing gut microbiota, chronic constipation, or CKD-related dietary interventions.

4. Although the authors mention participants could consume sake lees in various ways, it may help to standardize or at least detail these methods to reduce variability and better interpret outcomes.

By incorporating these recommendations—particularly adding a control group, clarifying the open-label design’s limitations, performing a thorough statistical justification for sample size, and further exploring confounding variables— will significantly enhance the scientific rigor and clarity of the manuscript.

Reviewer #2: This manuscript by Tokumaru et al investigates the effect of sake lees on the health of subjects suffering from chronic constipation. The authors performed a single-arm clinical trial to see the effect of sake lees consumption on fecal uremic toxin levels, plasma D-alanine, gut microbiota composition, and constipation severity. Eight subjects consumed 25g and 50g of sake lees every day for six weeks. Fecal indole and p-cresol levels initially fluctuated, but at the end of the trial, p-cresol levels were lower. The plasma D-alanine level, a potentially renoprotective component of sake lees, significantly increased, especially with the higher dose, while D-serine did not change. Significant improvement in constipation was observed, with improvements in CSS and quality-of-life scores seen as early as week two. There was an increased proportion of Firmicutes, which may indicate possible gut health benefits regarding the gut microbiome. However, no significant change was observed in alpha diversity and some toxin-producing bacteria. I have several concerns regarding this manuscript

1. This study lacks a control group. The absence of a control group limits the ability to compare the effects of sake lees against a placebo, reducing the robustness of causal inferences.

2. The study included only eight participants, which limits the statistical power and generalizability of the findings. Small sample sizes also increase the risk of outliers disproportionately influencing results.

3. The study duration is short. The six-week intervention may not have been long enough to observe sustained changes in gut microbiota composition or lasting health benefits.

4. The reliance on self-reported metrics, such as the Constipation Scoring System (CSS) and quality-of-life questionnaires, introduces the potential for bias and variability in participant responses.

5. Variables such as diet, physical activity, hydration levels, and other lifestyle factors that could influence constipation and gut microbiota were not controlled or systematically analyzed.

6. PLOS authors have the option to publish the peer review history of their article (what does this mean? ). If published, this will include your full peer review and any attached files.

**Do you want your identity to be public for this peer review?** For information about this choice, including consent withdrawal, please see our Privacy Policy .

Reviewer #1: **Yes: ** Simna Saraswathi Prasannakumari

Reviewer #2: No

---

## [Author Response · Author response to Decision Letter 1]

1 Feb 2025

Dear reviewer and editor,

Thank you for your review.

The points we have revised in response to your review are listed in the file “Response to the reviewer”. We would be grateful if you could check this.

Toshiaki Tokumaru

---

## [Decision Letter · Decision Letter 1]

18 Mar 2025

PONE-D-24-55416R1Effects of sake lees intake on fecal uremic toxins, plasma D-alanine, constipation, and gut microbiome in healthy adults: A single-arm clinical trialPLOS ONE

Dear Dr. Tokumaru,

Thank you for submitting your manuscript to PLOS ONE. After careful consideration, we feel that it has merit but does not fully meet PLOS ONE’s publication criteria as it currently stands. Therefore, we invite you to submit a revised version of the manuscript that addresses the points raised during the review process.

A rebuttal letter that responds to each point raised by the academic editor and reviewer(s). You should upload this letter as a separate file labeled 'Response to Reviewers'.A marked-up copy of your manuscript that highlights changes made to the original version. You should upload this as a separate file labeled 'Revised Manuscript with Track Changes'.An unmarked version of your revised paper without tracked changes. You should upload this as a separate file labeled 'Manuscript'

We look forward to receiving your revised manuscript.

Kind regards,

Sayed Haidar Abbas Raza

Academic Editor

PLOS ONE

Reviewers' comments:

Reviewer's Responses to Questions

**Comments to the Author**

1. If the authors have adequately addressed your comments raised in a previous round of review and you feel that this manuscript is now acceptable for publication, you may indicate that here to bypass the “Comments to the Author” section, enter your conflict of interest statement in the “Confidential to Editor” section, and submit your "Accept" recommendation.

Reviewer #1: All comments have been addressed

Reviewer #2: All comments have been addressed

Reviewer #3: (No Response)

2. Is the manuscript technically sound, and do the data support the conclusions?

Reviewer #1: Yes

Reviewer #2: Yes

Reviewer #3: Partly

3. Has the statistical analysis been performed appropriately and rigorously? 

Reviewer #1: Yes

Reviewer #2: Yes

Reviewer #3: Yes

4. Have the authors made all data underlying the findings in their manuscript fully available?

Reviewer #1: Yes

Reviewer #2: Yes

Reviewer #3: (No Response)

5. Is the manuscript presented in an intelligible fashion and written in standard English?

Reviewer #1: Yes

Reviewer #2: Yes

Reviewer #3: Yes

6. Review Comments to the Author

Reviewer #1: (No Response)

Reviewer #2: The authors have addressed all the comments. They have provided point by point answers to all the comments.

Reviewer #3: Dear Editor,

I write to submit my review comments on the manuscript titled “Effects of sake lees intake on fecal uremic toxins, plasma D-alanine, constipation, and gut microbiome in healthy adults: A single-arm clinical trial”

The authors conducted a pilot study using a single-arm trial without a control group to explore the relationship between sake lees intake and changes in fecal uremic toxin levels among individuals with constipation.

Comments

1) The authors employed a single-arm trial without a control group. Even though some researchers feel it is not mandatory, a comparison or control group is often beneficial in pilot studies to help assess the feasibility, effectiveness, and robustness of the study design and protocols before a larger, full-scale study. In addition, a control group allows researchers to assess whether the proposed intervention or method is likely to be effective and practical. It is possible to observe a change in a single-arm trial, but it becomes extremely difficult to gauge the likely impact of the proposed intervention (sake lees intake) without a control group during the pilot study in a full-scale study. The implication of doing a single arm without a control group must be thoroughly discussed.

2) Since there was no control group, the conclusion drawn from the pilot study must be interpreted with caution, and this must be clearly stated.

3) The authors indicated that it was an exploratory study, which was why no control group was studied, but added that the study will lay the groundwork for a clinical trial targeting chronic kidney disease (CKD) patients. This is the more reason why control patients should have been included: if the intervention (sake lee intake) is potentially harmful or has unknown risks, a control group can help ensure that participants in the main study are not exposed to unnecessary risks.

4) Authors stated that sample size was not performed due to the absence of prior data on effect size. However, the authors could have adopted the approach of Cohen’s et al (Cohen classified effect sizes as small (d = 0.2), medium (d = 0.5), and large (d ≥ 0.8) to simulate different sample sizes taking into account the budget.

Cohen, J. (2016). Things I have learned (so far). American Psychological Association.

Not estimating sample size with the justification that there was no effect size from previous studies is problematic.

5) There was no rigorous power analysis, but the authors used a one-sample t-test to test whether the mean value of the primary outcome measure equals zero. How did the authors determine whether the study was powered enough to test that hypothesis? A similar comment applies to the use of repeated-measure ANOVA.

7. PLOS authors have the option to publish the peer review history of their article (what does this mean? ). If published, this will include your full peer review and any attached files.

**Do you want your identity to be public for this peer review?** For information about this choice, including consent withdrawal, please see our Privacy Policy .

Reviewer #1: No

Reviewer #2: No

Reviewer #3: No

---

## [Author Response · Author response to Decision Letter 2]

9 Apr 2025

Please refer to “Responses to Reviewers”.

---

## [Decision Letter · Decision Letter 2]

15 May 2025

Effects of sake lees intake on fecal uremic toxins, plasma D-alanine, constipation, and gut microbiome in healthy adults: A single-arm clinical trial

PONE-D-24-55416R2

Dear Dr. Tokumaru,

We’re pleased to inform you that your manuscript has been judged scientifically suitable for publication and will be formally accepted for publication once it meets all outstanding technical requirements.

Kind regards,

Sayed Haidar Abbas Raza

Academic Editor

PLOS ONE

Additional Editor Comments (optional):

Reviewers' comments:

Reviewer's Responses to Questions

**Comments to the Author**

1. If the authors have adequately addressed your comments raised in a previous round of review and you feel that this manuscript is now acceptable for publication, you may indicate that here to bypass the “Comments to the Author” section, enter your conflict of interest statement in the “Confidential to Editor” section, and submit your "Accept" recommendation.

Reviewer #3: All comments have been addressed

2. Is the manuscript technically sound, and do the data support the conclusions?

Reviewer #3: Yes

3. Has the statistical analysis been performed appropriately and rigorously? 

Reviewer #3: Yes

4. Have the authors made all data underlying the findings in their manuscript fully available?

Reviewer #3: (No Response)

5. Is the manuscript presented in an intelligible fashion and written in standard English?

Reviewer #3: Yes

6. Review Comments to the Author

Reviewer #3: The authors have made conscious efforts to address all the limitations in the paper and where nothing can be done about it, they have admitted that as a limitation of the study and have clearly indicated it as such

7. PLOS authors have the option to publish the peer review history of their article (what does this mean? ). If published, this will include your full peer review and any attached files.

**Do you want your identity to be public for this peer review?** For information about this choice, including consent withdrawal, please see our Privacy Policy .

Reviewer #3: No

---

## [Editor Report · Acceptance letter]

PONE-D-24-55416R2

PLOS ONE

Dear Dr. Tokumaru,

I'm pleased to inform you that your manuscript has been deemed suitable for publication in PLOS ONE. Congratulations! Your manuscript is now being handed over to our production team.

Kind regards,

on behalf of

Dr. Sayed Haidar Abbas Raza

Academic Editor

PLOS ONE